# Phytostimulatory Influence of *Comamonas testosteroni* and Silver Nanoparticles on *Linum usitatissimum* L. under Salinity Stress

**DOI:** 10.3390/plants10040790

**Published:** 2021-04-16

**Authors:** Ahlam Khalofah, Mona Kilany, Hussein Migdadi

**Affiliations:** 1Biology Department, Faculty of Science, King Khalid University, P.O. Box 9004, Abha 61413, Saudi Arabia; 2Research Center for Advanced Materials Science (RCAMS), King Khalid University, P.O. Box 9004, Abha 61413, Saudi Arabia; monak@kku.edu.sa; 3Department of Microbiology, National Organization for Drug Control and Research (NODCAR), Cairo 12611, Egypt; 4Department of Plant Production, College of Food and Agriculture Sciences, King Saud University, Riyadh 11461, Saudi Arabia; 5National Agricultural Research Center, Baqa 19381, Jordan

**Keywords:** nano-silver, sodium chloride, salt stress, antioxidant, phenols, enzymes

## Abstract

They were shifting in land use increases salinity stress, significant abiotic stress affecting plant growth, limiting crop productivity. This work aimed to improve *Linum usitatissimum* L. (linseed) growth under salinity using *Comamonas testosteroni* and silver nanoparticles (AgNPs). AgNPs were fabricated exploiting *Rosmarinus officinalis* and monitored by U.V./Vis spectrophotometry scanning electron microscopy (SEM) and Fourier transforms infrared spectroscopy (FTIR). Photosynthetic pigments, enzymatic and nonenzymatic antioxidants of linseed were investigated under salt stress in treated and untreated plants with *C. testosteroni* alongside AgNPs. Our findings recorded the formation of AgNPs at 457 nm, which were globular and with a diameter of 75 nm. Notably, chlorophyll-a, b, and total chlorophyll reduction while enhanced carotenoids and anthocyanin contents were attained under salinity stress. Total dissoluble sugars, proline, and dissoluble proteins, H_2_O_2_, malondialdehyde, enzymatic and nonenzymatic antioxidants were significantly elevated in NaCl well. Combined AgNPs and *C. testosteroni* elevated photosynthetic pigments. Also, they led to the mounting of soluble sugars, proline, and soluble proteins. H_2_O_2_ and malondialdehyde decreased while enzymatic and nonenzymatic antioxidants increased in response to AgNPs, *C. testosteroni,* and their combination. Thus, AgNPs and *C. testosteroni* might bio-fertilizers to improve linseed crop productivity under salinity stress.

## 1. Introduction

The utmost serious environmental threat for plant survival and harvest yield is soil salinity. It affects 19.5% of inundated land and 2.1% of dry ground cultivation over the globe. Salinity poses several undesirable consequences for plants through hypertonic and hyperosmotic effects on several plant bio-processes, prompting membrane disorganization, increment in reactive oxygen species (ROS) levels, and metabolic harmfulness [1]. Salinity influences carotenoids, anthocyanin, chlorophyll content, soluble sugar, and lipid contents [2]. Under salt pressure, plants have created complex techniques to contend with these oxidative stresses using different antioxidants′ synchronous actions. Of these, catalase (CAT), superoxide dismutase (SOD), and peroxidase (POD) assume critical roles in detoxifying reactive oxygen species (ROS). Catalase and peroxidase are engaged together, converting H_2_O_2_ into oxygen and water [3]. Khan et al. [4] stated that plants adjust to osmotic stress by amassing some convenient solutes such as glycine betaine (G.B.), proline, trehalose, and polyols during salt pressure proline conduct a principal function in defending plants from osmotic pressure. Subsequently, antioxidants and convenient solutes may award a strategy to upgrade salt resistance in plants. Linseed (*Linum usitatissimum* L.) is an herbaceous plant all over the globe related to the *Linaceae* family. It has a broad scope of manufacturing uses because of its primary products, such as seeds and fibers [5]. Linseed is a significant nutritional crop in antioxidants and omega-3 fatty acids. Polyphenols are critical substances concerning the antioxidant features of the plants. Flavonoids serve as scavengers of different oxidizing free radicals [4]. Notwithstanding, the vulnerability of the yield to salt pressure is the major factor for diminished crop productivity. The survey is meager on the resistance of linseed plants to saltiness. Certain methods would be an effective strategy to improve harvest productivity [6]. It is highly desirable to alleviate salt stress′s adverse actions to fulfill the population′s increase globally. Chemical treatments and agronomical crop handling pursuit were tried to alleviate the salinity stress with little success [7]. A viable alternative is to induce plants′ capability to face the detrimental situation successfully by remediation using rhizosphere bacteria and AgNPs, which were accounted for relieving the ominous effects of salinity by improving the growth and yield of the plant. In an ever-changing world, nanoscience is a fascinating field of sciences, enabling the plants to endure salts influencing the plant system for ameliorating the plant growth and the potential of ROS scavenging. The green fabricated nanoparticles from plants are frugal and naturally benevolent [8]. Applications of nanoparticles can help limit the utilization of poisonous, brutal, and costly synthetic compounds used in the common processes of plant output [9]. Metal nanoparticles exert a significant action on plant growth that concerns food quality [10]. Oppositely, sometimes, AgNPs exert counteractive effects on crops [11]. It was widely reported that plants exposed to nanoparticles could use and translocate nanoparticles to different plant parts. Among nanoparticles, AgNPs have miscellaneous applications and have been extremely used as antimicrobial agents in cosmetics, household items, filters, and cosmetic items [12]. AgNPs boosted or diminished the plant development and biomass, relying on the dose, size, and exposure period [13]. As of late, hardly any investigations have detailed the positive function of AgNPs under saltiness [14]. Interestingly, AgNPs boost the seed up-growth of tomatoes; notwithstanding, little data is accessible concerning the impact of AgNPs on wheat seedlings in salty environments [13]. It was theorized that seed preparation with AgNPs might reduce the salt pressure in wheat plants by diminishing the oxidative stress by altering antioxidant enzyme activities leaning on the dosages of AgNPs applied [15]. Nowadays, many reports discussed the effects of AgNPs on improving the development and seed germination of plants like *Panicum vulgatum*, *Phytolacca Americana*, *Brassica juncea*, *Zea mays*, *Phaseolus vulgaris*, *Pennisetum glaucum*, *Boswellia ovalifoliolata* [10,16,17]. Since bacteria plentifully take part in the rhizosphere microorganisms, it is profoundly likely that they affected plant physiology, especially considering their association in root colonization [18]. Rhizobacteria enhancing plant outgrowth are soil-borne, free-living microbes, which boost plant growth and development directly or indirectly [19]. Microorganisms such as *Azospirillum*, *Xanthomonas*, *Pseudomonas*, *Alcaligenes faecalis*, *Rhizobium*, *Bradyrhizobium japonicum*, and *Acetobacter diazotrophicus* have been appeared to deliver auxins associate with invigorating plant growth [20]. *C. testosteroni* is a facultative anaerobic bacterium and selected in this study to alleviate the salinity threatening plant growth. It was previously used as a good bio-fertilizer [21]. *C. testosteroni* can also degrade phenol and 4-chlorophenol mixtures completely through a meta-cleavage pathway, which is beneficial not only for enhanced cell growth but also for the biotreatment of both compounds [22]. *C. testosteroni* is capitalized on in heavy metal bioremediation because of its high heavy metal tolerance [23]. The exploitation of PGPR offers an alluring method to supplant compost pesticides. PGPR is a part of coordinated management frameworks in which diminished paces of agrochemicals and cultural control rehearses biocontrol operators. Such an incorporated framework could be used for moving vegetables to create more fiery transfers that would be lenient to nematodes and different infections for at any rate hardly a few weeks in the wake of relocating to the field [24]. This work intends to assess AgNPs and rhizosphere bacteria′s influence in mitigation saltiness for *Linum usitatissimum* L. In Addition, monitoring of the physiological state of the plant was done that was represented by photosynthetic pigments, soluble protein, total soluble sugars, proline, hydrogen peroxidation (H_2_O_2_), malondialdehyde (MDA), total phenolics (TPC), glutathione (GSH), ascorbic acid (AsA), and the activity of the antioxidant enzymes (superoxide dismutase, catalase, peroxidase, ascorbate peroxidase, and glutathione reductase).

## 2. Results

### 2.1. Characterization of AgNPs

*Rosmarinus officinalis* was used in AgNPs generation. Color variation and spectroscopic analysis (Figure 1) are good evidence for Nano-silver biosynthesis, where a distinctive plasmon absorption peak was monitored at 475 nm.

As certified in Appendix A, FT-IR spectroscopy revealed the characteristic peaks of bio- compounds responsible for reducing and capping AgNPs. The apparent peaks at 3570–3050, 2926–2856, 1720, 1602, 1602, 1436, 1340, 1269, 1174, 982, and 690–540 cm^−1^ were related to the specific functional groups. Distinctive strong broadband of alcoholic compounds in the range ~3570–3050 cm^−1^ due to extending O-H groups′ vibration. There are two strong bands at 2926–2856 cm^−1^ because of extending the C-H bond′s vibration corresponding to methylene. Prominent peaks at 1720 and 1602 cm^−1^ could be attributed to extending the vibration of C=O that is allocated to aldehyde or ketone. Medium peaks emerged at 1602–1436–1340–1269 cm^−1^ attributed to stretching vibration of C=C bond referred to the aromatic compound. The weak band at 1174 cm^−1^ might be because of the stretching vibration of the aliphatic ether. The medium peak at 1030 cm^−1^ might be because of the stretching vibration of S=O assigned to sulfoxide. At 982 cm^−1^, a strong C=C bending vibration could be ascribed to a monosubstituted alkene. Strong peaks at 690–540 cm^−1^ may result from the stretching halo compound.

### 2.2. Photosynthetic Pigments

The findings showed a significant lowering in chlorophyll-a, b, and total chlorophyll at 50 and 100 mM NaCl. While total chlorophyll and chlorophyll-b were non-significantly decreased at 25 mM compared to the control as depicted in Table 1, however, the carotenoids and anthocyanin content significantly increased at 50 and 100 mM compared to the control as depicted (Table 1). Interestingly, AgNPs boost all pigments contents, particularly at 50 mM leading to a significant increase in total chlorophyll, chlorophyll-b, and carotenoids pigments. AgNPs led to significantly prompting in the total chlorophyll, carotenoids, and anthocyanin at 100 mM salt. There is no difference between total pigments from control plants and plants treated with *C. testosteroni*. Using both AgNPs and *C. testosteroni* offers to ascend to increase the photosynthetic pigment levels. The letter mixture caused a significant increase in chlorophyll-b, the total chlorophyll, carotenoids, and anthocyanin levels at 25 mM, while chlorophyll-b, the total chlorophyll, and carotenoids significantly increase at 50; meanwhile, all chlorophyll types reached the maximum levels at 100 mM.

### 2.3. Soluble Sugars, Proteins, and Proline Contents

Our results showed a significant boost of total soluble sugars, soluble proteins, and proline with an increase in salt concentration contrasted with the control. The generation of soluble sugars, soluble proteins, and proline was substantially higher with treatment with AgNPs and *C. testosteroni* combination than those at the use of AgNPs alone, while the lowest value was observed at the use of *C. testosteroni* alone in non-stressful and stressful linseed plants (Table 2).

### 2.4. MDA and H_2_O_2_ Contents

As per our outcomes, H_2_O_2_ content significantly increased by increasing salinity levels, and the most increased values were at 50- and 100-mM NaCl contrasted with control. Both H_2_O_2_ and MDA decreased in the stressed and non-stressed linseed plants after treatment with AgNPs, *C. testosteroni,* and the combination of both of them (Table 2).

### 2.5. Determination of Nonenzymatic Antioxidants

Our findings showed that TPC, AsA, and GSH significantly increased under salinity. No significant differences were found in TPC content, while significant increases in AsA and GSH were attained after applying AgNPs alone, *C. testosteroni* alone, and blending of them (Table 3).

### 2.6. Assay of Antioxidant Enzyme Activities

Data illustrated in Table 4 cleared that the progressive increase in NaCl concentrations (25, 50, 100 mM) resulted in a significant increase in the activity of antioxidant enzymes SOD, CAT, POD, APX, and G.R. contrasted with the control. Also, enzyme activity increased in the stressed and non-stressed linseed plants because of the treatment using AgNPs alone, *C. testosteroni,* and their combination.

## 3. Discussion

*C. testosteroni* was selected to alleviate the salinity threatening plant growth because it was previously used as a salinity challenger up to 3% concentration [25]. A color change successfully approved the greenly AgNPs fabrication into dark brown. The U.V.–visible spectroscopy empowered us to gauge the distinctive localized surface plasmon resonance peak and its maximum absorbance at more or less 425 nm correlated to the globular shaped AgNPs [26]. Notably, bio-compounds existing in rosemary plant extract associated with reducing AgNO_3_ resulting AgNPs were checked by FTIR spectroscopy. By matching the FTIR spectra of the rosemary plant extract and the bio-fabricated AgNPs, it could be noted that all peaks attained in plant extract were also observed in the FTIR spectrum of AgNPs, but shifted to higher-frequency positions. These shifts evidenced the functional groups assigned to these peaks were used for the bio-reduction and the stabilization of the resultant AgNPs. SEM micrograph illustrated the shape and size of AgNPs that was confirmed by the spherical shape of AgNPs with a size 75 nm. This result is consistent with other research [27]. Photosynthetic pigments including chlorophyll-a, chlorophyll-b, the total chlorophyll, carotenoids, and anthocyanin play a vital role in photosynthesis. However, salinity affects photosynthesis by reducing stomatal conductance by changing its water status, pigments′ concentration and altering the chloroplast ultrastructure [28]. The results imply that chlorophyll reduction because of salinity led to a lower photosynthesis rate, while AgNPs and *C. testosteroni* could improve plant development controllers or increase plant nutrient uptake [29]. A decline in chlorophyll might be because of the repression of accountable enzymes for its synthesis [30]. Pigment content suppression was attributed to upgraded chloroplast structure harm, pigment instability, and chlorophyllase production [6]. Nanoparticles have both favorable and unfavorable effects on seed germination, root elongation, cell division, chromosomal aberration, and metabolic activities [31]. However, Gupta et al. [10] confirmed the phytostimulatory impact of green-synthesized AgNPs during rice *(Oryza sativa* L.) seedling. The initiation of respiration and rapid ATP creation by the effect of AgNPs lowered the germination time and fasting seed germination [32]. Anyhow, the increment in photosynthetic pigments may be because of Na uptake minimizing by plants because of AgNPs [33]. Contrary, Thiruvengadam et al. [34] reported that AgNPs exerted no significant effect at 1.0 mg/L on total chlorophyll, whereas higher concentrations (5.0 and 10.0 mg/L) of AgNPs resulted in significantly decreased total chlorophyll. AgNPs have phytotoxic effects in some seedlings diminishing plant pigments [11]. It was hypothesized that ZnO-NPs involved in the rise of plant chlorophyll-and exceedingly efficient in the synthesis of photosynthetic pigments that increased photosynthesis rate [35]. It might be because nanoparticles are powerful amplifiers of photosynthetic effectiveness that in parallel cause light absorption by chlorophyll, as it causes convey of energy to nanoparticles from chlorophyll [36]. As observed in this work, the higher contents of carotenoids and anthocyanins formed by AgNPs exposure might be because of higher oxidative stress caused by AgNPs. Similar results have been found regarding anthocyanin levels in *A. thaliana* exposed to AgNPs [37]. Osmotic stresses, such as salinity, light, pH, and temperature, considerably affect anthocyanin′s stability [38]. Anthocyanins are associated with chlorophyll′s photoprotection and in response to osmotic stresses, for example, salinity in plants [39]. Sharma et al. [40] concluded that anthocyanins preserve the plant from visible light under salinity stress. Plant growth advancing rhizobacteria is considered the best remediation tool for plants under saline stress by diverse mechanisms [41]. As we observed, *C. testosteroni* have a significant part in salt tolerance and enhancing the development of *Linum usitatissimum* L. Comparable outcomes were obtained by Razzaghi et al. [42], who stated that the stimulatory effect of salinity–tolerant bacteria could be ascribed to improved mineral, nitrogen, and water uptake. Likewise, Zameer et al. [43] stated that plant development advancing rhizobacteria proved best for improving chlorophyll-a, chlorophyll-b, carotenoid, and anthocyanin of NaCl stressed tomato [43]. It is well known that proline and total soluble sugars are important natural solutes that keep the cell homeostasis and assist in cell osmoregulation under salinity stress and their aggregation in plants correlates with enhanced salt resilience. Soluble proteins are an important tool in osmoregulation under saline stress, providing plant cells with nitrogen and protecting them from potential oxidative damage [33]. Likewise, salt pressure enhanced soluble proteins, total soluble sugars, and production in chickpea and proline contents in wheat [44] and in *Brassica juncea* L. [45]. Besides, Mohamed et al. [46] have documented that AgNPs increased the organic solute concentrations in wheat seedlings under salt pressure, achieved the same findings. Also, Nano-zinc oxide boosts the organic solutes in lupine and tomato plants under salt stress through the activation of translational and/or transcriptional processes [7,47]. Oppositely, Nano-cerium oxide nanoparticles decreased proline contents in leaves of *B. napus* under salinity [14]. The utilization of beneficial bacteria mitigating stress discovering alternative approaches involved in stress tolerance [48]. Similarly, it was announced that treatment with *B. subtilis* and *Arthrobacter* sp. boost proline, total soluble sugars, and soluble proteins [49]. Salinity drives to hyperosmotic stress and ionic imbalance, hence, induce reactive oxygen species, including H_2_O_2_, which could extremely harm the lipids, photosynthetic pigments, nucleic acids proteins, and cell membranes [50]. Concurrently, similar results were approximately shown regarding lipid peroxidation (MDA), which substantially increased for stressed plants at all salt concentrations. Concurrently, these results are concordant with several studies expressed that the content of H_2_O_2_ and MDA were significantly elevated in different plants under salinity where high aggregation of MDA may be attributed to the destruction of the cellular membrane probity, and cellular compounds, like proteins and lipids [33]. AgNPs reduced H_2_O_2_ and MDA contents, thus improving the injury normally induced by salinity stress. AgNPs have been accounted for upregulating the antioxidant system by speedy disposal of H_2_O_2_, subsequently prompting growth development upkeep [51], which is consistent with the outcomes of Burman et al. [52], who reported that zinc nanoparticles induced defensive effects on biomembranes versus alternations of membrane permeability and oxidative stress in chickpea seedlings [33]. Gupta et al. [10] realized a similar outcome that investigated that AgNPs decreased MDA and H_2_O_2_ content. *Pseudomonas* spp. were effective in salinity tolerance by constringing H2O2 content [53]. Nonenzymatic antioxidants (TPC, AsA, & GSH) are involved in many cellular processes either by playing critical roles in plant tolerance or acting as enzyme cofactors, affecting plant prosperity and development from initial development phases senescence [54]. Glutathione is a powerful reducing agent, which plays an important role in eliminating ROS either as an individual molecule or through the ascorbate–glutathione cycle [55]. The manifest accumulation of nonenzymatic antioxidants in the lupine plant because of nanoparticles may help salt resistance via osmotic change; improve plant prosperity [7]. Nanoscience has become a fundamental and emerging tool in agronomy for inducing crop production by suppressing disease factors [56]. Our results are consistent with others who declared that by elevating the intensity of NaCl, total phenolic content significantly increases [44,57]. Our finding was consistent with another investigation where AgNPs, *Acinetobacter* sp. and *Bacillus* sp. promoted phenolic compounds and anti-oxidation potential because of depletion of free radicals. GSH and AsA were promptly increased with an increase in the concentration of biosynthesized AgNPs in *T. foenum-graecum,* while the lowermost intensity of GSH and AsA was found in *Z. mays* and *A. cepa* L. seedlings [58]. It has been thought that increasing the antioxidant enzyme levels under saltiness is an effective strategy to confer salt resistance. Notably, phenolic compounds played a vital function in safeguarding the plants against harmful effects induced by various pressures like salinity [45]. On contrary, it was stated that GSH and AsA contents of safflower significantly decreased with rising salinization levels. However, bacterial strains *B. cereus*, and *B. aerius* elevated the level of ascorbic acid and glutathione in safflower seedlings with elevation in salinity level [59]. Antioxidants enzymes are the utmost common physiological factors regulating plant growth, and they are effective in scavenging ROS through their increased activity under abiotic tension [60]. Similar to our results, several types of research concerning NaCl- treated plants have documented that enzymes′ activity of APX, SOD, CAT, and G.R. increased in *Solanum lycopersicum*, *Brassica juncea* L., *Zea mays*, and *Pistachio vera* in response to a progressive elevation in NaCl [61]. It was documented that the seedlings of mango rootstock exhibited greater CAT, SOD enzyme activities than control plants under salinity [62]. In close effects of AgNPs are predictable with different outcomes who revealed an essential increase in the rate of seed germination during treatment with AgNPs in *B. ovalifoliolata*, *Z. mays* L., *A. cepa* L., and *T. foenum-graecum* L. [58]. Consistently, it actually proved the increase in CAT, SOD, AXP, and POD activities because of AgNPs [63]. It has been investigated the high production of CAT and POD that diminished the ROS release and Nano-toxicity and the odds of oxidative pressure in plants [64]. Interestingly, microorganisms could assume a huge part in the management of saltiness stress [65]. *C. testosteroni* reportedly has favorable effects on plant growth, higher yield, and abiotic tolerance. The selected bacterial strain maintained a higher growth rate under zinc stress (unpublished results from author′s lab). Because of salinity and Zn abiotic pressure resistance, siderophore and organic acids can prompt supplement bioavailability and improving soil aggregation. Thus, *C. testosteroni* is considered a superior strain in resisting the negative effects of NaCl. In consonance with [3] Habib et al., a substantial increase of antioxidant enzyme activity (APX, CAT, GR, POD, and SOD) occurred under saline conditions in the okra plants treated with PGPR [3]. Identically, other studies stated that *B. cereus* inoculation significantly increased the antioxidant enzymes (POD, SOD, and CAT) activities that of great importance to cope with oxidative stress during salt stress conditions [66]. This shows the effectiveness of enzymatic activities in controlling the possible oxidative damage under bacterial inoculation [67]. Oppositely, sometimes AgNPs exerted unfavorable effects on antioxidant enzymes [11]. The elevation in the activity of H_2_O_2_ scavenging enzymes like CAT and POD allows us to speculate that H_2_O_2_ homeostasis has altered in inoculated plants, which led to increased plant prosperity under pressure. It was reported that beneficial bacteria could increase the enzyme activity for ROS scavenging because of their action on genes encoding for antioxidant enzymes [68]. Similar results were reported for *Solanum tuberosum* and *Lactuca sativa* treated with *Bacillus strains* and *Pseudomonas mendocin,* respectively [69]. Peroxidase enzyme is associated not only in scavenging H_2_O_2_ but also in plant prosperity, development, suberization, lignification, and crosslinking of cell wall compounds that prevent the entry of more ions [70]. These results recommend that *C. testosteroni* and AgNPs can be used in salinized agricultural farming grounds as a bio-inoculant to prompt crop productivity.

## 4. Materials and Methods

### 4.1. Bacterial Culture Preparation

Rhizosphere bacteria *C. testosteroni* has been isolated from the soil rhizosphere, Aseer region, KSA. It was cultured in a nutrient broth medium and incubated at 150 rpm for 48 h at 27 °C. Afterward, the cells were gathered by centrifugation at 5000 rpm for 15 min, rinsed, and resuspended in sterilized water to a concentration of nearly 1 × 10^7^ CFU mL^−1^. For soil application, nutrient-free bacterial suspension was sprayed, and the control pots were sprayed with a similar volume of sterile water [71].

### 4.2. Preparation of AgNPs Using Rosmarinus Officinalis

Chemicals (AgNO_3_) of pure grade were used (Merck, Ltd., Feltham, U.K.). The mature leaves of rosemary were collected from Abha City, Saudi Arabia, rinsed using distilled water, air-dried at 20–25 °C for five days, then ground into a coarse powder. About 10 g of the powder were suspended in 100 mL distilled water at 25 °C for 24 h, filtered through a muslin cloth to discard the fibers, and then filtered through Whatman filter paper (No1). Ultimately, it was centrifuged at 5000 rpm for 10 min to separate the clear leaf extract that will be preserved at 4 °C until used as a reducing and stabilizer agent in nanoparticle preparation [26]. An aqueous solution of AgNO3 (1 mM) was added drop-wise into 50 mL of rosemary leaf extract. The mixture was incubated for 18 h at room temperature. Control without AgNO3 was also kept at the same conditions. The solution was centrifuged for 10 min at 10,000 rpm to isolate the AgNPs. The nanoparticles were washed several times using deionized water and then suspended in 95% ethanol before characterization.

### 4.3. Description of Bio Fabricated AgNPs

The color alteration was examined within 24 h that potentially showed the development of AgNPs. Characterizations of the bio-formed AgNPs were studied via U.V.–Vis spectroscopy analysis using UV-3600 Shimadzu spectrophotometer (Duisburg, Germany), and the manufacturing of AgNPs was monitored within the range (200–600 nm). Fourier transform infrared spectroscopy (FT-IR) was done using Perkin Elmer Spectrum 2000 (Waltham, MA, USA), at a rate of 16 times within the range 600–4000 cm^−1^, and clarity of 4 cm^−1^. The shape and size of the produced AgNPs were depicted by a scanning electron microscope (SEM, JEM-1011, JEOL, Tokyo, Japan) at a quickening voltage of 90 kV.

### 4.4. Effect of AgNPs and C. testosteroni on Plant Growth

Linseeds used in this study were got from the ministry of agriculture, Abha, Saudi Arabia. First, several NaCl concentrations (0, 25, 50, 100 mM) were prepared using distilled water. Foremost, surface-sterilization of seeds occurred by steeping in 70% ethanol for 5 min, then in 2% sodium hypochlorite for 30 min. Next, sterile seeds were washed, sterilized, and sown in 15-cm plastic pots containing equal amounts of sand and peat moss. The seeds were dispersed at a profundity of 1 cm at 20–25 °C in a greenhouse enlightened with regular light. All pots were watered with 200 mL water trice a week; then they were divided into four groups; each group contained five replicates per treatment (250 seeds per transaction) and was stressed by various NaCl salt concentrations (0, 25, 50, 100 mM). Each group was partitioned into four groups, where the first group was left with no treatment as a control, the second group was foliar sprayed with AgNPs solution, the third group comprised soil sprayed with a bacterial suspension of *C. testosteroni*. The fourth group included seedlings sprayed with both AgNPs and a nutrient-free bacterial suspension of *C. testosteroni*. The treatment protocol was carried out for 21 days, and then the plants were washed with sterile distilled water, rinsed, and prone to physiological analysis [28].

### 4.5. Quantification of Photosynthetic Pigments

The chlorophyll content was determined by homogenizing 0.2 g of mushy leaves in 80% chilled acetone (10 mL) in the dark, then was carried out using 100% acetone [72]. Carotenoid content and anthocyanin pigment were assessed according to Afroz et al. [2].

### 4.6. Determination of Non-Antioxidant Enzymes

#### 4.6.1. Determination of Soluble Sugars, Soluble Protein, and Proline

A 0.2 g of mushy leaves were homogenized in 10 mL 96% ethanol (*v/v*) then washed by 5 mL 70% ethanol (*v/v*). Subsequently, the prepared extract was centrifuged for 10 min at 3500 g, and the supernatant was put away at 4 °C for measurement. Total soluble sugar concentrations were determined by boiling 3 mL of freshly prepared anthrone reagent (100 mL of 72% sulfuric acid (*v/v*) containing 150 mg anthrone) with 0.1 mL of the alcohol extract for 10 min. After cooling, the absorbance was recorded at 625 nm using a spectrophotometer (UV-1900 BMS, Duisburg, Germany) to gauge total soluble sugars′ quantity using a glucose standard curve [73]. For total soluble protein assay, about 0.5 g of fresh-ground leaves were well homogenized in phosphate buffer (0.05 M-pH 7.8) under cooling, filtered, and centrifuged for 10 min 12,000× *g* at 4 °C. Uv-Vis spectrum was noted at 595 nm [74]. A 0.5 g of green leaves were soaked in 5 mL of 3% sulfosalicylic acid. About 2 mL of 1% ninhydrin (*w/v*) in 60% acetic acid (*v/v*), 20% ethanol (*v/v*) were mixed with the plant extract and boiled in a water bath at 100 °C for 30 min. After cooling, 6 mL of toluene was added to a separate chromophore gauged at 520 nm [75].

#### 4.6.2. Determination of Hydrogen Peroxide (H_2_O_2_) and Malondialdehyde (MDA)

H_2_O_2_ content of plant leaves was calorimetrically measured as Mukherjee and Choudhuri [76]. Aliquot of 200 μL acetone extract was mixed with 0.04 mL of 0.1% TiO_2_ and 0.2 mL NH_4_OH (20%). The pellet was decollated with acetone and resuspended in 0.8 mL H_2_SO_4_. The mixture was then centrifuged at 6000 g for 15 min, and the supernatant was read at 415 nm. The MDA determination showed by Wu estimated lipid peroxidation level [77]. The frozen specimens were crushed in fluid nitrogen and then extracted in 5 mL of 0.5% thiobarbituric acid (TBA), which disintegrated in 5% trichloroacetic acid (TCA). The extract was boiled for 40 min, immediately cooled, and centrifuged for 10 min at 12,000 g. Spectral analysis was carried out at 523 nm. An extinction coefficient of 155 mM L^−1^ cm^−1^ was used to determine lipid peroxidation level.

#### 4.6.3. Determination of Total Phenolic (TPC), Ascorbic Acid (AsA), and Glutathione (GSH)

Approximately 100 µL plant extract was combined with Folin–Ciocalteu reagent (0.75 mL) at 22 °C for 5 min. Then, 0.75 mL of Na_2_CO_3_ was incubated with the mixture at 22 °C for 90 min. The absorbance was monitored at 725 nm [78]. Ascorbic acid was determined according to [79]. About 1g of plant leaves was homogenized immediately in fluid nitrogen, extracted with 10 mL 5% (*w/v*) trichloroacetic acid (TCA), and then centrifuged at 4 °C for 5 min at 15,000 g. The supernatant was immediately investigated for AsA content in a 1 mL reaction mixture containing 50 µL 10 mM dithiothreitol (DDT), 100 µL 0.2 M phosphate buffer (pH 7.4), 0.5% (*v/v*) N-ethylmaleimide, 10% (*w/v*) trichloroacetic acid (TCA), 42% (*v/v*) H_3_PO_4_, 4% (*v/v*) 2,2′-Diphyridyl, and 3% (*w/v*) FeCl_3_. The spectral analysis was estimated at 525 nm. GSH was estimated, according to Anderson [80], where nearly 0.5 g mushy leaves were macerated in 2 mL 5% sulfosalicylic acid undercooling and centrifuged at 12000 g for 10 min. A mixture of 0.5 mL supernatant, 0.6 mL of phosphate buffer (100 mM), pH 7, and 40 µL of 5′5′-dithiobis-2-nitrobenzoic acid (DNTB) was prepared. After 2 min, the absorbance was observed at 412 nm.

### 4.7. Determination of Antioxidant Enzymes

Almost 0.5 g of green leaves were ground, homogenized, filtered, and centrifuged at 12,000 g for 10 min at 4 °C [81]. Superoxide dismutase (SOD) was determined following Li et al. [82]. Potassium phosphate buffer (50 mM, pH 7.8), 13 mM methionine, 75 µL NBT, 2 µL riboflavin, 0.1 mM EDTA was mixed with 100 µL of plant extract. The units of enzymes exploited to inhibit the reduction of 50% of the nitro blue tetrazolium (NBT) represent one unit of SOD activity, which was estimated at 560 nm. To determine Catalase (CAT) A reaction mixture composed of 100 µL plant extract, 100 µL 0.3 M H_2_O_2_, 2 mM EDTA and 2.8 mL phosphate buffer (0.050 M, pH 7). The CAT activity was estimated using the formula (ε = 39.4 mM^−1^cm^−1^) by checking the absorbance decline at 240 nm because of H_2_O_2_ disappearance [83].

Assay of Peroxidase POD activity was carried out as per Zhou and Leul [84]. A mixture of 50 mM potassium phosphate buffer (pH 7), 0.4% H_2_O_2_, 1% guaiacol, and 100 µL enzyme extract was subjected to spectral analysis to monitor the variation absorbance because of guaiacol at 470 nm. Ascorbate peroxidase (APX) was assayed as Nakano and Asada [85]. A mixture of 100 µL enzyme extract, 2.7 mL potassium phosphate buffer (25 mM), 100 µL H_2_O_2_ (300 mM), 100 µL ascorbate (7.5 mM), and 2 mM EDTA (pH 7) were mixed well. The alteration detected the oxidation of ascorbate in absorbance at 290 nm (ε = 2.8 mM^−1^cm^−1^). The glutathione reductase (G.R.) enzyme activity was measured after trice monitoring the state of oxidation of NADPH taken at 340 nm and activity expressed as ∆ A340 min^−1^ mg^−1^ protein [86].

### 4.8. Statistical Analysis

Data were subjected to analysis using two-way analysis of variance (ANOVA), and the honestly significant difference (HSD) at *p* < 0.05 probability level using Tukey post hoc test to compare the differences among treatment means using SAS software (version 9.1 Institute, Cary, NC, USA) [87].

## 5. Conclusions

It is deduced that *C. testosteroni* and silver nanoparticles are jointly involved in ameliorating tolerance of linseed to salt stress. This manuscript reports a unique study describing the positive effects of AgNPs on linseed growth under salinity. *C. testosteroni* and silver nanoparticles′ synergistic interaction stimulated the production of photosynthetic pigments, sugars, proteins, proline, and antioxidants, whether enzymatic or nonenzymatic, lowered the contents of H_2_O_2_ and MDA. Thus, AgNPs combined with *C. testosteroni* might cultivate linseed plants and energize plants′ growth and economic yield growing in highly salted soils.

## Figures and Tables

**Figure 1 plants-10-00790-f001:**
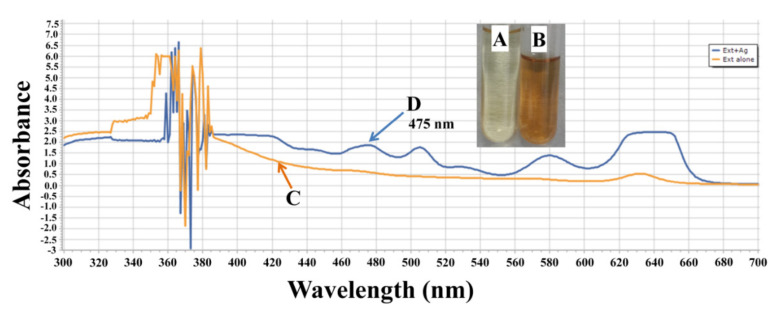
Color change and UV-Visible spectral analysis of *Rosmarinus officinalis* plant extract and synthesized AgNPs. Where (**A**) denotes a plant extract, (**B**) denotes plant extract, and AgNO_3_, (**C**) denotes spectral analysis of plant extract, and (**D**) denotes spectral analysis of synthesized AgNPs.

**Table 1 plants-10-00790-t001:** Impact of AgNPs, *C. testosteroni,* and their combination on chlorophyll-a, chlorophyll-b, total chlorophyll, carotenoids, and anthocyanin pigments of *Linum usitatissimum* L. plants under salinity stress (0, 25, 50, and 100 mM NaCl). Mean ± Sd values for treatment over three replications. According to Tukey′s test, different letters within the same columns show significant differences (*p* < 0.05). The higher cases are the differences among salinity treatments, and the lower cases are the differences among AgNPs+C.t treatments′ mean. ns: means not significant.

Salinity Levels/Treatments		Chlorophyll-a (mg/g)	Chlorophyll-b (mg/g)	Total Chlorophyll (mg/g)	Carotenoids (mg/g)	Anthocyanin (mg/g)
0.0 mM NaCl	Control	1.27 ± 0.089	0.97 ± 0.082	2.06 ± 0.008	0.63 ± 0.033	0.31 ± 0.024
	AgNPs	1.32 ± 0.065	1.02 ± 0.036	2.34 ± 0.003	0.68 ± 0.045	0.35 ± 0.169
	C.t	1.29 ± 0.031	0.99 ± 0.024	2.28 ± 0.006	0.65 ± 0.061	0.33 ± 0.143
	AgNPs + C. t	1.38 ± 0.022	1.05 ± 0.017	2.43 ± 0.001	0.73 ± 0.183	0.39 ± 0.008
NaCl treatment Mean		1.32 ± 0.052 A	1.01 ± 0.04 A	2.28 ± 0.005 A	0.67 ± 0.08 D	0.35 ± 0.087 C
25 mM NaCl	Control	1.19 ± 0.034	0.91 ± 0.046	2.10 ± 0.012	0.69 ± 0.026	0.39 ± 0.014
	AgNPs	1.23 ± 0.068	0.97 ± 0.023	2.20 ± 0.067	0.74 ± 0.003	0.47 ± 0.157
	C.t.	1.21 ± 0.047	0.93 ± 0.054	2.14 ± 0.005	0.71 ± 0.117	0.43 ± 0.082
	AgNPs + C. t	1.25 ± 0.021	0.99 ± 0.091	2.24 ± 0.002	0.78 ± 0.093	0.51 ± 0.163
NaCl treatment Mean		1.23 ± 0.043 B	0.95 ± 0.038 A	2.17 ± 0.022 A	0.73 ± 0.06 C	0.45 ± 0.104 B
50 mM NaCl	Control	1.12 ± 0.040	0.82 ± 0.076	1.94 ± 0.008	0.74 ± 0.085	0.42 ± 0.079
	AgNPs	1.16 ± 0.079	0.89 ± 0.090	2.05 ± 0.036	0.80 ± 0.131	0.46 ± 0.181
	C.t.	1.14 ± 0.022	0.84 ± 0.054	1.98 ± 0.004	0.76 ± 0.060	0.44 ± 0.003
	AgNPs + C. t	1.18 ± 0.011	0.93 ± 0.027	2.11 ± 0.073	0.83 ± 0.007	0.49 ± 0.001
NaCl treatment Mean		1.15 ± 0.038 B	0.87 ± 0.050 B	2.02 ± 0.03 B	0.78 ± 0.07 B	0.45 ± 0.07 B
100 mM NaCl	Control	0.98 ± 0.045	0.63 ± 0.033	1.61 ± 0.021	0.82 ± 0.012	0.45 ± 0.086
	AgNPs	1.06 ± 0.037	0.69 ± 0.025	1.75 ± 0.009	0.85 ± 0.078	0.51 ± 0.191
	C.t.	1.02 ± 0.081	0.67 ± 0.091	1.69 ± 0.016	0.83 ± 0.164	0.48 ± 0.054
	AgNPs + C. t	1.10 ± 0.034	0.72 ± 0.086	1.82 ± 0.042	0.87 ± 0.051	0.53 ± 0.008
NaCl treatment Mean		1.04 ± 0.05 C	0.68 ± 0.06 C	1.72 ± 0.022 C	0.84 ± 0.08 A	0.49 ± 0.085 A
(Ag/Ct) treatment Mean						
Control		1.14 ± 0.052 c	0.83 ± 0.060 d	1.93 ± 0.012 d	0.72 ± 0.041 d	0.39 ± 0.050 d
AgNPs		1.19 ± 0.062 b	0.89 ± 0.044 b	2.08 ± 0.028 b	0.77 ± 0.064 b	0.45 ± 0.174 b
C. t		1.17 ± 0.045 b	0.86 ± 0.056 c	2.02 ± 0.008 c	0.74 ± 0.101 c	0.42 ± 0.071 c
AgNPs + C. t		1.23 ± 0.022 a	0.92 ± 0.055 a	2.15 ± 0.029 a	0.81 ± 0.084 a	0.48 ± 0.045 a
An honestly significant difference (HSD) at *p* < 0.05 probability level using Tukey′s test for:						
NaCl treatments		0.075	0.052	0.126	0.059	0.035
(Ag/Ct) treatment		0.033	0.023	0.056	0.025	0.015
NaCl × Ag/C. t Interaction		ns	ns	ns	ns	ns

**Table 2 plants-10-00790-t002:** Impact of AgNPs, *C. testosteroni* and their combination on soluble sugar, soluble proteins, proline, hydrogen peroxide, and lipid peroxidation of *Linum usitatissimum* L. plants under salinity stress (0, 25, 50, and 100 mM NaCl). Mean ± Sd values for treatment over three replications. According to Tukey′s test, different letters within the same columns show significant differences (*p* < 0.05). The higher cases are the differences among salinity treatments, and the lower cases are the differences among AgNPs + C.t treatments′ mean. ns: means not significant.

Salinity Levels/Treatments		Soluble Sugar mg/g	Soluble Proteins mg/g	Proline mg/g	Hydrogen Peroxide µg/L	Lipid Peroxidation µg/L
0.0 mM NaCl	Control	121.56 ± 1.394	24.87 ± 1.851	10.23 ± 0.174	2.58 ± 0.045	20.87 ± 0.021
	AgNPs	127.22 ± 1.271	27.39 ± 1.664	15.19 ± 0.153	1.61 ± 0.005	19.33 ± 0.065
	C. t	124.61 ± 1.382	25.01 ± 1.041	12.68 ± 0.12	1.81 ± 0.079	20.01 ± 0.028
	AgNPs + Ct	131.80 ± 1.190	30.44 ± 1.009	17.45 ± 0.18	1.98 ± 0.012	19.82 ± 0.061
NaCl treatment Mean		126.3 ± 1.31 B	26.93 ± 1.40 D	13.89 ± 0.16 D	2.0 ± 0.035 D	19.93 ± 0.044 D
25 mM NaCl	Control	128.64 ± 1.006	28.03 ± 1.061	13.97 ± 0.120	4.04 ± 0.004	23.88 ± 0.011
	AgNPs	134.21 ± 1.037	31.68 ± 1.043	17.99 ± 0.104	3.72 ± 0.091	21.91 ± 0.083
	C. t	131.99 ± 1.982	28.99 ± 1.927	14.05 ± 0.145	3.96 ± 0.067	22.06 ± 0.003
	AgNPs + Ct	137.01 ± 1.003	35.05 ± 1.082	20.74 ± 0.191	3.89 ± 0.043	22.01 ± 0.092
NaCl treatment Mean		131.61 ± 1.257 AB	30.94 ± 1.28 C	16.69 ± 0.14 C	3.90 ± 0.051 C	22.47 ± 0.047 C
50 mM NaCl	Control	131.85 ± 1.481	31.40 ± 1.049	19.62 ± 0.157	5.35 ± 0.0028	29.65 ± 0.006
	AgNPs	136.42 ± 1.031	34.26 ± 1.003	23.08 ± 0.140	4.92 ± 0.001	27.34 ± 0.073
	C. t	133.97 ± 1.156	32.41 ± 1.017	22.49 ± 0.128	5.12 ± 0.035	28.51 ± 0.039
	AgNPs + Ct	137.39 ± 1.294	38.98 ± 1.050	26.73 ± 0.123	5.03 ± 0.009	28.93 ± 0.020
NaCl treatment Mean		134.91 ± 1.241 A	34.26 ± 1.03 B	22.98 ± 0.137 B	4.65 ± 0.012 B	28.61 ± 0.035 B
100 mM NaCl	Control	137.79 ± 1.003	35.29 ± 1.001	22.89 ± 0.182	7.19 ± 0.015	35.40 ± 0.031
	AgNPs	141.23 ± 1.932	39.58 ± 1.054	26.65 ± 0.115	6.80 ± 0.070	34.21 ± 0.002
	C. t	139.95 ± 1.096	36.99 ± 1.082	23.04 ± 0.171	6.93 ± 0.041	34.76 ± 0.007
	AgNPs + C. t	143.11 ± 1.800	42.17 ± 1.038	29.95 ± 0.185	6.90 ± 0.082	35.02 ± 0.059
NaCl treatment Mean		140.52 ± 1.458 A	38.51 ± 1.04 A	25.63 ± 0.163 AA	6.96 ± 0.052 A	34.85 ± 0.025 A
(Ag/ Ct) treatment Mean						
Control		129.96 ± 1.221 c	29.9 ± 1.24 d	16.68 ± 0.16 d	4.79 ± 0.02 a	27.45 ± 0.02 a
AgNPs		134.77 ± 1.32 ab	33.23 ± 1.19 b	20.73 ± 0.13 b	4.26 ± 0.04 c	25.7 ± 0.06 b
C.t		132.63 ± 1.40 bc	30.85 ± 1.27 c	18.07 ± 0.14 c	4.46 ± 0.06 bc	26.36 ± 0.02 ab
AgNPs + C. t		137.33 ± 1.32 a	36.66 ± 1.04 a	23.72 ± 0.17 a	4.45 ± 0.04 b	26.45 ± 0.06 a
An honestly significant difference (HSD) at *p* < 0.05 probability level using Tukey′s test for:						
NaCl treatments		9.71	2.79	1.99	0.482	0.502
(Ag/Ct) treatment Mean		4.14	1.21	0.853	0.188	0.015
NaCl × Ag/C. t Interaction		ns	ns	2.4; 3.6*	0.53; 2.42	ns

* Comparisons of means for the same level of salt (HSD = 2.4) and different levels of salt (HSD = 3.6).

**Table 3 plants-10-00790-t003:** Impact of AgNPs, *C. testosteroni,* and their combination on total phenolics (TPC), ascorbic acid (AsA), and glutathione (GSH) of *Linum usitatissimum* L. plants under salinity stress (0, 25, 50, and 100 mM NaCl). Mean ± Sd values for treatment over three replications. According to Tukey′s test, different letters within the same columns show significant differences (*p* < 0.05). The higher cases are the differences among salinity treatments, and the lower cases are the differences among AgNPs + C.t treatments′ mean. ns: means not significant.

Salinity/AgNPs + Ct Treatments		TPC mg/g	AsA mg/g	GSH nM/g
0.0 mM NaCl	Control	5.24 ± 0.115	7.66 ± 0.005	360.34 ± 0.176
	AgNPs	5.61 ± 0.102	12.04 ± 0.062	386.98 ± 0.139
	C. t	5.38 ± 0.176	10.92 ± 0.017	379.01 ± 0.023
	AgNPs + C.t	5.57 ± 0.192	9.97 ± 0.043	380.43 ± 0.105
NaCl treatment Mean		5.45 ± 0.156 D	10.15 ± 0.032 D	376.69 ± 0.11 C
25 mM NaCl	Control	8.12 ± 0.171	11.84 ± 0.098	371.92 ± 0.199
	AgNPs	9.03 ± 0.173	14.73 ± 0.002	390.06 ± 0.132
	C. t	8.72 ± 0.105	12.06 ± 0.083	386.55 ± 0.101
	AgNPs + C. t	8.50 ± 0.183	12.87 ± 0.063	388.72 ± 0.102
NaCl treatment Mean		8.59 ± 0.158 C	12.88 ± 0.062 C	384.31 ± 0.133 BC
50 mM NaCl	Control	11.01 ± 0.197	15.25 ± 0.065	386.59 ± 0.011
	AgNPs	11.37 ± 0.131	17.44 ± 0.0188	406.31 ± 0.162
	C. t	11.25 ± 0.122	16.91 ± 0.024	399.89 ± 0.190
	AgNPs + C. t	11.32 ± 0.134	17.11 ± 0.029	401.57 ± 0.122
NaCl treatment Mean		11.24 ± 0.146 B	16.68 ± 0.034 B	398.59 ± 0.121 AB
100 mM NaCl	Control	13.06 ± 0.165	19.06 ± 0.088	409.36 ± 0.145
	AgNPs	13.28 ± 0.193	20.82 ± 0.005	416.80 ± 0.057
	C.t	13.09 ± 0.126	19.75 ± 0.005	410.58 ± 0.168
	AgNPs + C. t	13.15 ± 0.104	20.09 ± 0.061	412.11 ± 0.102
NaCl treatment Mean		13.145 ± 0.147 A	19.93 ± 0.04 A	412.21 ± 0.12 A
(Ag/C. t) treatment Mean				
Control		9.36 ± 0.172 b	13.45 ± 0.064 c	382.05 ± 0.133 b
AgNPs		9.82 ± 0.15 a	16.26 ± 0.022 a	400.04 ± 0.122 a
C.t		9.61 ± 0.132 ab	14.91 ± 0.032 b	394.01 ± 0.12 a
AgNPs + C. t		9.64 ± 0.153 a	15.01 ± 0.049 b	395.71 ± 0.11 a
An honestly significant difference (HSD) at *p* < 0.05 probability level using Tukey′s test for:				
NaCl treatments		0.93	1.42	28.39
(Ag/C. t) treatment Mean		0.36	0.56	11.97
NaCl × Ag/C. t Interaction		ns	1.60; 2.51*	ns

* Comparisons of means for the same level of salt (HSD = 1.6) and different levels of salt (HSD = 2.51).

**Table 4 plants-10-00790-t004:** Impact of AgNPs, *C. testosteroni,* and their combination on superoxide dismutase (SOD), catalase (CAT), Peroxidase POD, Ascorbate peroxidase APX, and glutathione reductase (G.R) of *Linum usitatissimum* L. plants under salinity stress (0, 25, 50, and 100 mM NaCl). Mean ± Sd values for treatment over three replications. According to Tukey′s test, different letters within the same columns show significant differences (*p* < 0.05). The higher cases are the differences among salinity treatments, and the lower cases are the differences among AgNPs + C.t treatments′ mean. ns: means not significant.

Salinity /AgNPs + C. t Treatments		SOD U/mg Protein	CAT U/mg Protein	POD U/mg Protein	APX U/mg Protein	G.R U/mg Protein
0.0 mM NaCl	Control	103.42 ± 0.199	205.64 ± 0.191	101.26 ± 0.153	96.97 ± 0.124	120.53 ± 0.112
	AgNPs	129.50 ± 0.112	220.87 ± 0.113	134.56 ± 0.189	118.26 ± 0.157	145.37 ± 0.117
	C. t	116.39 ± 0.117	217.02 ± 0.081	117.92 ± 0.127	105.67 ± 0.129	132.65 ± 0.106
	AgNPs + C.t	121.05 ± 0.143	209.63 ± 0.175	126.55 ± 0.130	112.89 ± 0.108	129.80 ± 0.187
NaCl treatment Mean		117.59 ± 0.13 C	213.29 ± 0.14 B	120.073 ± 0.150 C	108.45 ± 0.13 C	132.088 ± 0.131 C
25 mM NaCl	Control	119.10 ± 0.156	210.75 ± 0.118	113.68 ± 0.107	104.69 ± 0.113	126.43 ± 0.144
	AgNPs	132.89 ± 0.103	232.69 ± 0.137	153.54 ± 0.114	130.62 ± 0.194	157.26 ± 0.156
	C. t	127.75 ± 0.107	224.55 ± 0.122	129.72 ± 0.172	124.17 ± 0.198	140.05 ± 0.107
	AgNPs + C. t	129.64 ± 0.119	217.22 ± 0.119	134.89 ± 0.183	118.45 ± 0.176	146.33 ± 0.159
NaCl treatment Mean		127.3 ± 0.121 C	221.31 ± 0.124 B	132.96 ± 0.144 AB	119.48 ± 0.17 B	142.52 ± 0.142 B
50 mM NaCl	Control	125.93 ± 0.130	235.10 ± 0.116	118.51 ± 0.145	113.89 ± 0.151	132.64 ± 0.169
	AgNPs	154.80 ± 0.127	257.29 ± 0.132	136.92 ± 0129	128.86 ± 0.126	167.39 ± 0.126
	C. t	141.61 ± 0.109	243.77 ± 0.069	121.87 ± 0.136	119.31 ± 0.023	151.67 ± 0.141
	AgNPs + C. t	148.78 ± 0.111	248.99 ± 0.185	126.12 ± 0.118	123.11 ± 0.160	162.90 ± 0.101
NaCl treatment Mean		142.78 ± 120 B	246.23 ± 0.125 A	125.86 ± 0.132 BC	121.29 ± 0.115 B	153.65 ± 0.134 A
100 mM NaCl	Control	133.30 ± 0.106	249.67 ± 0.168	126.63 ± 0.112	116.59 ± 0.127	146.51 ± 0.124
	AgNPs	168.88 ± 0.159	263.86 ± 0.172	147.95 ± 0.190	148.94 ± 0.004	158.10 ± 0.116
	C. t	156.31 ± 0.197	251.84 ± 0.015	135.16 ± 0.135	123.66 ± 0.139	149.14 ± 0.110
	AgNPs + C. t	161.07 ± 0.120	257.62 ± 0.04	139.74 ± 0.080	135.07 ± 0.028	153.25 ± 0.108
NaCl treatment Mean		154.85 ± 0.145 A	255.75 ± 0.098 A	137.37 ± 0.129 A	131.07 ± 0.075 A	151.75 ± 0.115 A
(Ag/C. t) treatment Mean						
Control		120.44 ± 0.150 b	225.29 ± 0.148 c	120.44 ± 0.152 d	108.04 ± 0.13 d	131.53 ± 0.14 d
AgNPs		146.48 ± 0.125 b	243.66 ± 0.139 a	143.24 ± 0.16 a	131.67 ± 0.12 a	157.03 ± 0.144 a
C. t		135.5 ± 0.133 a	234.278 ± 0.072 b	126.17 ± 0.143 c	118.203 ± 0.122 c	143.38 ± 0.116 c
AgNPs + C. t		140.11 ± 0.123 a	233.34 ± 0.130 b	131.83 ± 0.128 b	122.38 ± 0.118 b	148.07 ± 0.139 b
An honestly significant difference (HSD) at *p* < 0.05 probability level using Tukey′s test for:						
NaCl treatments		11.19	17.97	9.39	9.27	10.86
(Ag/Ct) treatment Mean		4.63	7.45	4.06	3.94	4.57
NaCl × Ag/C. t Interaction		13.13; 20.04*	ns	11.51; 17.04	11.17; 16.74	3.07; 19.53

* Comparisons of means for the same level of salt (HSD = 13.13) and different levels of salt (HSD = 20.04).

## Data Availability

The data presented in this study are available on request from the corresponding author.

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
