# Peer review of "Phytostimulatory Influence of Comamonas testosteroni and Silver Nanoparticles on Linum usitatissimum L. under Salinity Stress"

_plants, 2021, doi:10.3390/plants10040790_

Round 1

Reviewer 1 Report

The manuscript need to be carefully rechecked and intensively, where

  1. The title is fine but I suggest adding Salt stress as keyword.
  2. I think that results from Soluble sugars, proteins, and proline are not contents but concentrations. Revise the text and figures.
  3. The difference between the mean comparison analysis method recommended Duncan test.
  4. Please follow the regulations drawn form J. Plants.

Minor correction in English needs to be checked “especially the use of articles”.

P1, line 3: L => L.

P1, line 27: H2O2 => H2O2

P2, line 82: Zea mays => Zea mays

P3, line 108: AgNPs. => AgNPs

P4, line 136, 137: total chlorophyll- => total chlorophyll

P8, line 221: Oryza sativa L's seedling => rice (Oryza sativa L.) seedlings

P8, line 261: Arthrobacter sp => Arthrobacter sp.

P9, line 277: Pseudomonas spp => Pseudomonas spp.

P9, line 289: Acinetobacter sp. and Bacillus sp. => Acinetobacter sp. and Bacillus sp.

P9, line 304: Pistachio =>Pistacia vera

P9, line 308: A. cepa L => A. cepa L.

P9, line 313, 317: C testosterone => C. testosterone

P9, line 321: B cereus => B. cereus

P10, line 337: preparation. => preparation

P10, line 344: Rosmarinus officinalis. => Rosmarinus officinalis

P10, line 362: C testosterone => C. testosterone

P11, line 383: antioxidant enzymes. => antioxidant enzymes

P11, line 384: 4.6.1. Determination of soluble sugars, soluble protein, and proline. => 4.6.1. Determination of soluble sugars, soluble protein, and proline

P12, line 425: antioxidant enzymes. => antioxidant enzymes

P12, line 446: Statistical analysis. => Statistical analysis

P13, line 507: (Solanum lycopersicum) => (Solanum lycopersicum)

P15, line 600: (Solanum lycopersicum Mill.) => (Solanum lycopersicum Mill.)

Author Response

Dear Professor

Thank you very much for your valuable comments and suggestions. We considered every point raised and corrected it in a different color. 

For your kind attention, we attached a file response to reviewers.

Best regards

Reviewer 2 Report

The manuscript “Phytostimulatory influence of Comamonas testosteroni and Silver nanoparticles on Linum usitatissimum L under salinity stress”, is well executed what concerns experimental design, however, the statistic analysis applied to analyze the data is not the most appropriated one, in the opinion of this reviewer.

Sometimes, “too much is less” and this reviewer thinks that some of the sentences used to describe the work are confusing and not entirely correct. Simplifying the English in those cases, would benefit the comprehension of the work herein presented. For instances, in page 4 line 144, “The content of all pigments non significantly increased when treated with C. testosteroni.” Statistically speaking there is no difference between total pigments from control plants and plants treated with C. testosteroni, therefore these results just show a trend.      

Major Concerns

After carefully reading the manuscript, this reviewer has a major concern regarding the statistical analysis applied. This reviewer this the describe statistical analysis is not the most appropriated one and that might lead the authors to discussions and conclusions that might not be entirely correct. Please, find below the suggestions to correct the issue.

Page 12 – In 4.8. Statistical analysis, this reviewer thinks the analysis performed on the results is not the most adequate considering the experimental design described by the authors.

  • A two-way ANOVA should be used due to different treatments applied to the plants – different salt concentrations and different soil amendments ( testoteroni, AgNPs and C. testoteroni+AgNPs)
  • Instead of Fisher’s LSD test, the authors should perform a Dunnett test or a Tukey test; however maybe the Dunnett test would be the most appropriate since the authors are trying to compared every means to the mean of the control plants (no salt, no bacterium, no AgNPs). The Dunnett test also accounts for multiple comparisons. The Fisher’s LSD testis not the best test since it does not correct for multiple comparisons.

Other concerns

After carefully reading the manuscript, this reviewer has questions/suggestions that should be revised and addressed by the authors.

Material and Methods

Page 10 – In 4.2. Preparation of AgNPs using Rosmarinus officialis, please, provide a list of the chemicals used in the reactions. A brief description of the synthesis would be helpful. 

Results

  • Page 3 - Figure 1 represents the spectroscopic analysis of AgNPs biosynthesis by the plant rosemary (Rosmarinus officinalis) – it should be stated in the Figure legend which plant.
  • Page 4 – However, legends in Figures 2 and 3 claims that theAgNPs were biosynthesized by Linum usitatissimum. Assuming that Figure 1 has the corrected plant from where the AgNPS were synthesyzed, the authors should correct the sentences.
  • Figure 2. – Represents the FT-RT spectra of R. officinalis after the addition of AgNO3 of is unclear; the values in the background should be taken out – or provided as supplementary information. Additionally, the if y axis represents a percentage of transmittance, the symbol (%) should be add. Finally, this reviewer would like to see the FT-IR spectra of the leaf extract of rosemary before the addition of AgNO3 and after addition of AgNO3.
  • Figure 3. – The SEM micrograph does not clearly show the AgNPs. A better micrograph should be provided, and arrows should be added to point out the AgNPs.  
  • The statistic analysis should be redone for Figures 4, 5, 6, 7, and 8, according to the suggestions stated in “Major concerns” section of this review.

Author Response

Dear Professor

Greetings

Thank you very much for the valuable comments and suggestions that improved our work and add value to our manuscript. All comments are considered and were inserted in the text. For your kind attention, we attached a response to the reviewer comment file. 

Best regards

Round 2

Reviewer 2 Report

The authors of the manuscript “Phytostimulatory influence of Comamonas testosteroni and Silver nanoparticles on Linum usitatissimum L under salinity stress” did a good job addressing most of the concerns explained by this reviewer. The statistical analysis was completely redone, following the previous suggestions.

Minor concerns

Results

Here and there, this reviewer has still found some sentences that do not reflect results accurately.

The use on “non-significantly elevation or decrease”, means that there are no statistical differences between treatments. Therefore, when describing this type of results the authors should refer to a “trend” was observed or simply state that “no significant differences were found”. For example:

Pag. 8 – line 204: “Non-significantly elevation in TPC and significant elevation in AsA and GSH were attained…” should be replace with “No significant differences were found in TPC content, while significant increases in AsA and GSH were attained…”

The authors should check the entire manuscript carefully and change similar sentences accordingly.

Discussion

The discussion should be checked carefully, and changes should be made to reflect the new revised tables. For instances:

Pag. 12, line 263 – Figure 2 is now Supplementary Figure S2

Pag. 13, line 294 – There is not Figure 4 in the current version of the manuscript.

Pag. 14, line 346 – “Our finding was concurrence with…” should be replaced with “Our finding was consistent with…”

Pag. 14, line 355 – “…bacterial strains B. cereus, B. aerius…” should be replaced with “…bacterial strains B. cereus and B. aerius…”.

Author Response

Dear Professor

Thank you very much for the valuable comments you suggested which improved more our work. We are pleased to attach responses to the comments. All corrections were made in red color in the manuscript.

Best regards
